# *Rhizopus oryzae*-Mediated Green Synthesis of Magnesium Oxide Nanoparticles (MgO-NPs): A Promising Tool for Antimicrobial, Mosquitocidal Action, and Tanning Effluent Treatment

**DOI:** 10.3390/jof7050372

**Published:** 2021-05-10

**Authors:** Saad El-Din Hassan, Amr Fouda, Ebrahim Saied, Mohamed M. S. Farag, Ahmed M. Eid, Mohammed G. Barghoth, Mohamed A. Awad, Mohammed F. Hamza, Mohamed F. Awad

**Affiliations:** 1Department of Botany and Microbiology, Faculty of Science, Al-Azhar University, Nasr City, Cairo 11884, Egypt; hema_almassry2000@azhar.edu.eg (E.S.); mohamed.farag@azhar.edu.eg (M.M.S.F.); aeidmicrobiology@azhar.edu.eg (A.M.E.); mohamed_gamal.sci@azhar.edu.eg (M.G.B.); 2Department of Zoology and Entomology, Faculty of Science, Al-Azhar University, Nasr City, Cairo 11884, Egypt; Mohamed_awad@azhar.edu.eg; 3Guangxi Key Laboratory of Processing for Non-Ferrous Metals and Featured Materials, School of Resources, Environment and Materials, Guangxi University, Nanning 530004, China; m_fouda21@hotmail.com; 4Nuclear Materials Authority, El-Maadi, Cairo POB 530, Egypt; 5Department of Biology, College of Science, Taif University, P.O. Box 11099, Taif 21944, Saudi Arabia; m.fadl@tu.edu.sa; 6Botany and Microbiology Department, Faculty of Science, Al-Azhar University, Assiut Branch, Assiut 71524, Egypt

**Keywords:** MgO-NPs, optimization, antimicrobial, mosquitocidal and repellence activity, tannery effluents, chromium ion

## Abstract

The metabolites of the fungal strain *Rhizopus oryaze* were used as a biocatalyst for the green-synthesis of magnesium oxide nanoparticles (MgO-NPs). The production methodology was optimized to attain the maximum productivity as follows: 4 mM of precursor, at pH 8, incubation temperature of 35 °C, and reaction time of 36 h between metabolites and precursor. The as-formed MgO-NPs were characterized by UV-Vis spectroscopy, TEM, SEM-EDX, XRD, DLS, FT-IR, and XPS analyses. These analytical techniques proved to gain crystalline, homogenous, and well-dispersed spherical MgO-NPs with an average size of 20.38 ± 9.9 nm. The potentiality of MgO-NPs was dose- and time-dependent. The biogenic MgO-NPs was found to be a promising antimicrobial agent against the pathogens including *Staphylococcus aureus*, *Bacillus subtilis*, *Pseudomonas aeruginosa*, *Escherichia coli*, and *Candida albicans* with inhibition zones of 10.6 ± 0.4, 11.5 ± 0.5, 13.7 ± 0.5, 14.3 ± 0.7, and 14.7 ± 0.6 mm, respectively, at 200 μg mL^–1^. Moreover, MgO-NPs manifested larvicidal and adult repellence activity against *Culex pipiens* at very low concentrations. The highest decolorization percentages of tanning effluents were 95.6 ± 1.6% at 100 µg/ 100 mL after 180 min. At this condition, the physicochemical parameters of tannery effluents, including TSS, TDS, BOD, COD, and conductivity were reduced with percentages of 97.9%, 98.2%, 87.8%, 95.9%, and 97.3%, respectively. Moreover, the chromium ion was adsorbed with percentages of 98.2% at optimum experimental conditions.

## 1. Introduction

The revolution of nanoscience has been intensely grown daily. Nanoscience refers to the production of new materials at the nanoscale (1–100 nanometers). The prepared materials have unique properties that are not found in bulk materials [1]. Among these unique properties are shape, size, compatibility, surface charge, chemical stability, catalytic activity, and small size to a large surface area [2]. These properties enable them to integrate into various biotechnological and biomedical applications. The nanomaterial compounds have been fabricated using different routes including chemical, physical, and biological methods [3]. The physical methods were achieved under harsh working conditions and consumed high energy with complicated experimental devices [4], while the synthesis by chemical routes took place using deleterious organic solvents with dangerous reducing agents and produced undesirable by-products with negative impacts to the surrounding environment [5]. Therefore, the researchers’ attention is directed to biological approaches to overcome or avoid the disadvantages associated with physical and chemical methods.

Biological synthesis or green synthesis has been described as cost-effectiveness, biocompatible, eco-friendly nature, and scalable, avoiding harsh conditions and not utilizing hazardous chemicals [6]. Therefore, it can be incorporated into various applications such as textiles, wastewater treatment, paper preservation, the food industry, cosmetics and pharmaceuticals, optics, and smart devices [7,8,9,10,11]. Various biological entities such as bacteria, fungi, yeast, actinomycetes, and plant extracts are utilized in the green synthesis of different metal and metal oxide nanoparticles, such as Ag, Cu, CuO, ZnO, TiO_2_, Se, and Fe_2_O_3_ [12,13,14,15,16].

Among important metal oxide nanoparticles, magnesium oxide nanoparticles (MgO-NPs) are characterized by biocompatibility, high stability, cost-effectiveness, high ionic properties, and having a crystal structure and safe and high contaminant adsorbents [17]. Miscellaneous applications and unique properties of MgO-NPs has made it highly attractive to researchers everywhere during the past two decades over other metal oxide nanoparticles [18]. Chemically, MgO-NPs can be synthesized through numerous processes including solvent modification, co-precipitation, moist chemical, sol-gel, and hydrothermal, but these methods have drastic effects on the environment [19]. Recently, the biological synthesis of nanoparticles using fungi has become the preferable method because of the flexibility of handling, multiplication, high growth rates, species diversity (more than 1.52 million species), cost-effectiveness, novelty, and environmentally friendly nature [20,21]. The synthesis of MgO-NPs using fungi can be achieved either by intracellular method through the transportation of metal ions inside the fungal cell and then reduced by enzymes, or extracellular method through the reacting of metal ions with fungal biomass filtrate [22]. It has been reported that MgO-NPs can be used in novel applications including the removal of toxic waste, catalysis, antimicrobial property, refractory materials and wastewater treatment, ceramics, heavy fuel oils, enhancement of the potential of antioxidant and substrate in ferroelectric thin films, biomedical fields, sensing, adsorbents, lithium batteries, and agriculture sectors [18,23,24,25]. MgO-NPs are characterized by long-lasting antimicrobial activity, and this phenomenon can be attributed to its ability to tolerate high temperatures and low volatility [26].

The common house mosquito *Culex pipiens* L. is present in Egypt and plays a critical role in the transmission of different human pathogens such as malaria parasites, filarial worms, and viruses of Rift valley fever, West Nile, and Japanese encephalitis [27]. This insect acquired resistance against different insecticides in Egypt [28], and therefore it is necessary to urgently discover new alternative insecticides.

The current study aims to myco-synthesize MgO-NPs by harnessing metabolites of the fungal strain *Rhizopus oryaze* isolated from a soil sample. The characterization of myco-synthesized MgO-NPs was attained by UV-Vis spectroscopy and TEM, SEM-EDX, XRD, DLS, FT-IR, and XPS analyses. Moreover, factors including pH, contact times, incubation temperature, and precursor concentrations that affect the myco-synthesis of MgO-NPs were optimized. The biomedical activities of MgO-NPs including antibacterial, antifungal, larvicidal, and repellency activities were investigated. Finally, the efficacy of MgO-NPs in decolorization and chromium adsorption from tanning effluents were assessed.

## 2. Materials and Methods

### 2.1. Chemicals Used

In the current study, magnesium nitrate hexahydrate (Mg (NO_3_)_2_·6H_2_O) and sodium hydroxide (NaOH) were used as analytical grade and obtained from Sigma Aldrich, Cairo, Egypt. The Malt Extract agar (MEA) media used for fungal isolations and Mueller-Hinton agar media used for antimicrobial activities were ready-made (Oxoid, Cairo, Egypt). All biological reactions were carried out using distilled water (dis. H_2_O). The tannery wastewater was collected from Robbiki Leather City, 10th of Ramadan, Cairo, Egypt (GPS: 30°17′898” N, 31°76′840” E).

### 2.2. Isolation and Identification of the Fungal Strain

The fungal strain used for biosynthesis of MgO-NPs was isolated from Qalyubia Governorate, Egypt (31°18’522.07” E, 30°15’524.13” N), and has a code E3. The isolation procedure was achieved according to Fouda et al. [29] as follows: approximately 1.0 g of soil sample undergo diluted in sterilized dis. H_2_O. After that, 100 µL of the fourth dilution was inoculated onto MEA plates and incubated for 3–4 days at 30 ± 2 °C. The appeared colonies were picked up and re-inoculated onto the same media for purifications. The purified colony was preserved on an MEA slant for further use.

The identification was accomplished by cultural and microscopic characteristics and confirmed by molecular identification using internal transcribed spacer (ITS) sequence analysis. The ITS rDNA region was amplified using primers for ITS1 f (5-CTTGGTCATTTAGAGGAAGTAA-3) and ITS4 (5-TCCTCCGCTTATTGATATGC-3) [30]. The PCR mixture contained: 1X PCR buffer, 0.5 mM MgCl_2_, 2.5 U Taq DNA polymerase (QIAGEN, Germantown, MD 20874, USA), 0.25 mM dNTP, 0.5 µL of each primer, and 1 µg of extracted genomic DNA. The PCR was performed in a DNA Engine Thermal Cycler (PTC-200, BIO-RAD, Hercules, CA, USA) with a program of 94 °C for 3 min, followed by 30 cycles of 94 °C for 30 s, 55 °C for 30 s, and 72 °C for 1 min, followed by a final extension performed at 72 °C for 10 min. The PCR product was checked for the expected sizes on 1% agarose gel and was sequenced by Sigma Company for Scientific Research, Egypt, with the two primers. The sequence was compared against the GenBank database using the NCBI BLAST tool. Multiple sequence alignment was done using the Clustal Omega software package (http://www.clustal.org/clustal2, accessed on 17 November 2010), and a phylogenetic tree was constructed using the neighbor-joining method with MEGA (v.6.1) software, with confidence tested by bootstrap analysis (1000 repeats).

### 2.3. Green Synthesis of MgO-NPs

#### 2.3.1. Preparation of Fungal Biomass Filtrate

Three disks (0.8 cm in diameter) of the old culture of fungal strain E3 were inoculated into 100 mL of malt extract broth (MAB) media and incubated for 5 days at 30 ± 2 °C under 150 rpm shaking condition. At the end of the incubation period, the inoculated MAB was centrifuged and fungal biomass was collected. About 10.0 g of fungal biomass was resuspended in 100 mL dis. H_2_O for 48 h. at 30 ± 2 °C and 150 rpm shaking condition. The previous mixture was centrifuged at 10,000 rpm for 5 minutes and the supernatant (fungal biomass filtrate) was collected and used for green synthesis of MgO-NPs.

#### 2.3.2. Green Synthesis of MgO-NPs

Approximately 102.5 mg of Mg(NO_3_)_2_·6H_2_O was dissolved in 10 mL dis. H_2_O, then mixed with 90 mL of fungal biomass filtrate overnight to produce a final concentration of 4 mM. At the end of the incubation period, the Mg(OH)_2_ was formed as turbid white precipitate, which was collected and rinsed with dis. H_2_O to remove any impurities before being oven-dried at 100 °C for 1 h (Equation (1)).
(1)$$\mathrm{Mg}({\mathrm{NO}}_{3}{)}_{2}\cdot 6H_{2}O+H_{2}O\overset{\mathrm{Fungal}}{\underset{\mathrm{Metabolities}}{\to }}\mathrm{Mg}(\mathrm{OH}{)}_{2}$$

The formed Mg(OH)_2_ was calcinated at 400 °C for 3 h to form MgO-NPs, as represented in Equation (2) [31]. The controls including fungal biomass filtrate and Mg(NO_3_)_2_·6H_2_O solution ran alongside the experiment under the same conditions.
(2)$$\mathrm{Mg}(\mathrm{OH}{)}_{2}\;\overset{400{}^{\circ}C}{\to }\;\mathrm{MgO}$$

### 2.4. Optimizing Myco-Synthesis of MgO-NPs

The MgO-NPs production and distribution were affected by the environmental factors such as pH, contact times, incubation temperature, and precursor concentrations. These factors were investigated at maximum surface plasmon resonance (SPR) detected by using a UV-Vis spectrophotometer (Jenway 6305, Staffordshire, UK). The effect of pH values at 6, 7, 8, 9, 10, and 11 on MgO-NPs sorption properties was investigated. The contact time either between fungal biomass and distilled water to produce biomass filtrate (24, 48, 72, and 96 h) or times between biomass filtrate and precursors (6, 12, 24, 36, 48, and 72 h) were also investigated. The incubation temperature (25 °C to 40 °C with intervals of 5 degrees) and precursor concentrations (1–5 mM) were assessed. At the end of each parameter, 1.0 mL of the sample was withdrawn and measured at maximum SPR at λ_max_ = 282 nm.

### 2.5. Characterization of Biosynthesized MgO-NPs

The particle sizes and shapes of biosynthesized MgO-NPs were detected using Transmission Electron Microscopy (TEM) (JEOL 1010, Japan, acceleration voltage of 120 KV). A drop of NP solution was loaded on the carbon-copper grid and underwent vacuum desiccation for 24 h and placed after that onto a specimen holder [32]. The elemental compositions of biosynthesized MgO-NPs were assessed using scanning electron microscopy connected with energy dispersive X-ray (SEM-EDX) (JEOL, JSM-6360LA, Akishima, Japan). The crystallographic structure of biosynthesized MgO-NPs was determined using X-ray diffraction (XRD) analysis by X’Pert pro diffractometer (Philips, Eindhoven, Netherlands). The operating conditions were, 2θ values measured in ranges of 4° to 80°, X-ray radiation source was Ni-filtered Cu Ka and the operating voltage and current were 40 KV and 30 mA, respectively. The average NP sizes were measured using the Debye–Scherrer equation [33] as follows:(3)D=Kλ/βcosθ
where D is average particle size, K is the Scherrer’s constant (0.9), λ is the wavelength of X-ray radiation (0.154 nm), and β and θ are the half of maximum intensity and Bragg’s angle, respectively. Moreover, the size distribution of MgO-NPs in colloidal solution was investigated by dynamic light scattering (DLS) analysis. The sample was subjected to measurement by Zeta sizer nano series (Nano ZS, Malvern, UK).

On the other hand, the functional groups present in fungal biomass filtrate involved in reduction, capping, and stabilization of MgO-NPs were investigated using Fourier transform infrared (FT-IR) spectroscopy (Agilent system Cary 660 FT-IR model). The MgO-NPs sample was ground with KBR pellets (1% *w*/*w*), pressure was applied to form a disk, and scanning was done in the range of 400–4000 cm^−1^.

Finally, the X-ray photoelectron spectroscopy (XPS) analysis was analyzed by ESCALAB 250XI^+^ instrument (Thermo Fischer Scientific, Inc., Waltham, MA, USA) connected with monochromatic X-ray Al Kα radiation (1486.6 eV). The analysis was conducted under the following conditions: the size of the spot was 500 µm, the samples were prepared under a pressure adjusted at 10^−8^ mbar., the energy was calibrated with Ag3d_5/2_ signal (∆BE: 0.45 eV) and C 1s signal (∆BE: 0.82 eV), and the full and narrow-spectrum passed energies were 50 eV and 20 eV, respectively [34,35].

### 2.6. Antimicrobial Activity

The antimicrobial activity of MgO-NPs synthesized by fungal metabolites was investigated against various pathogenic microbes, including *Staphylococcus aureus* ATCC 6538, *Bacillus subtilis* ATCC 6633 (Gram-positive strains), *Pseudomonas aeruginosa* ATCC 9022, *Escherichia coli* ATCC 8739 (Gram-negative strains), and unicellular fungi *Candida albicans* ATCC 10231. The bacterial strains were subcultured on nutrient agar media (containing g L^–1^: peptone, 5; beef extract, 3; NaCl, 5; agar, 20; distilled water, 1000 mL) while *C. albicans* was subcultured on yeast extract peptone dextrose (YEPD) agar media (containing g L^–1^: glucose, 20; peptone, 20; yeast extract, 10; agar, 20; distilled water, 1000 mL) for 24 h. To check antimicrobial activity, each strain was homogenously streaked over Mueller-Hinton agar (for bacterial strains) and YEPD agar plates (for *C. albicans)* using a sterilized cotton swab. Three wells (0.7 cm diameter) were cut in the streaked Mueller-Hinton plates and filled with 100 µL of biosynthesized MgO-NPs (200 µg mL^–1^). The minimum inhibitory concentration (MIC) was assessed by using different concentrations of MgO-NPs (150, 100, 50, and 25 µg mL^–1^). The loaded Mueller-Hinton plates were kept in the refrigerator for 1 h before being incubated at 35 °C for 24 h. The results were recorded as a zone of inhibitions (ZOIs) around each well by mm [36]. The experiment was achieved in triplicate.

### 2.7. Mosquitocidal Bioassay

#### 2.7.1. Mosquito Rearing

*Culex pipiens* L. mosquitos’ vectors were reared in the Laboratory of Medical Entomology, Animal House, Department of Zoology and Entomology, Faculty of Science, Al-Azhar University, Cairo, Egypt. The mosquitoes were preserved at 30–35 °C and 60–80% relative humidity with a photoperiod of 12:12 h light: dark. Larvae were reared in white plastic cups (30 cm × 35 cm × 5 cm and containing 500 mL tap water). Fish was added to each tray for 2 weeks for optimum feeding of the larvae as follows: 0.1 g for 1st and 2nd instar larvae, 0.3 g for 3rd instar larvae, and 0.5 g for 4th instar larvae per day, until the pupation stage appeared. New pupae were transferred from the trays to plastic cups containing water and placed in screened cages (size 30 cm × 30 cm × 30 cm) until they emerged as adults. The adults were constantly provided with 5% sucrose solution on saturated cotton pads. The adult female mosquitoes of 5 days were destitute of sugar for 12 h, then fed on human blood by artificial membrane feeding technique [37] for 30 min. After two days of blood-feeding, plastic cups full of tap water were placed inside the cage for oviposition, which occurred on day 3 or 4 after blood feeding.

#### 2.7.2. Larvicidal Activity

The larvicidal activity (at room temperature) was assessed by the standard method of the World Health Organization (WHO) with slight modifications using the method described by Velayutham and Ramanibai. [38]. Twenty-five 3rd instar larvae of *C. pipiens* were moved separately from their colony in the laboratory to a plastic beaker (250 mL) containing 100 mL of the biogenic MgO-NPs (10 µg mL^−1^). The control was set up using chlorinated tap water. The same experiment was repeated with different concentrations of MgO-NPs (8, 6, 4, and 2 µg mL^–1^). The larval mortality percentages were counted every 24 h for 72 h using the following equation [39]:(4)$$\begin{array}{l} \mathrm{Mortality}\;\mathrm{percentages}(\%)=\\ \frac{\mathrm{mortality}\;\mathrm{percentages}\;\mathrm{of}\;\mathrm{treatment}-\mathrm{mortality}\;\mathrm{percentages}\;\mathrm{of}\;\mathrm{control}}{100-\mathrm{mortality}\;\mathrm{percentages}\;\mathrm{of}\;\mathrm{control}}\times 100 \end{array}$$

The lethal concentrations (LC_50_) and higher lethal concentrations (LC_90_) values were calculated at their 95% confidence intervals, as previously reported [40].

#### 2.7.3. Repellent Activity

The repellent study was conducted according to the method described by WHO [41]. This test was monitored by the Faculty of Science, Zoology and Entomology Department, Al-Azhar University, Cairo, Egypt, according to WHO ethics. Briefly, 3-day-old, blood-starved female *C. pipiens* were held in a net cage (45 cm × 30 cm × 45 cm). The volunteer had no contact with perfumes, lotions, perfumed soaps, or oils for one day before the experiment. From each volunteer arm, only 25 cm^2^ of skin, upper side, were uncovered and exposed to female *C. pipiens*, while the residual arm area was covered with elastic gloves. The biosynthesized MgO-NPs were applied at 1.0, 2.5, 5.0, and 10.0 mg/cm^2^ separately to the uncovered arm area. The commercial DEET (N, N-Diethyl-meta-toluamide) was used as a positive control. The repellent experiment was achieved during the night from 7:00 p.m. to 3:00 a.m. The volunteer introduces their arm (control and treated) simultaneously into the cage, gently tapping the sides of the experimental cages to activate the mosquitoes. The experiment was repeated three times for each concentration used. The volunteer was asked to insert their arm (control and treated) into the experiment cage simultaneously for 1 minute every 5 minutes. The mosquitoes that settled on the arm were recorded and then shaken off before sucking any blood. The percentages of repellency were calculated according to the following equation [42]:(5)$$\mathrm{Repellency}\;\mathrm{percentages}(\%)=\frac{T_{a}-T_{b}}{T_{a}}\times 100$$
where T_a_ and T_b_ denote the number of mosquitoes in the control and treated groups, respectively.

### 2.8. Bio-Adsorption and Treatment of Tannery Effluent

The decolorization of tanning effluents using biosynthesized MgO-NPs was investigated at different concentrations (50, 75, and 100 mg/100 mL) for different contact times (60, 120, 180, and 240 min). The experiment was conducted on a 250 mL conical flask containing 100 mL of tanning effluent mixed with specific NP concentrations. The mixture was stirred for 30 min before the experiment to reach absorption/desorption equilibrium. At the end of each incubation time, 1.0 mL of the mixture (tanning effluent with NPs) was withdrawn and centrifuged at 10,000 rpm for 3 min, and its optical density was measured at the maximum absorption band (λ_max_) of tanning effluent (550 nm) by a spectrophotometer (721 spectrophotometers, M-ETCAL). The decolorization percentages (%) of tanning effluents were calculated according to the following equation [10]:(6)$$\mathrm{Decolorization}\;\mathrm{percentages}(\%)=\frac{C_{0}-C_{t}}{C_{0}}\times 100$$
where C_0_ is the absorbance at zero time and C_t_ is the absorbance after specific time t (min).

Based on the most suitable MgO-NPs concentration at the optimum contact time, the biological oxygen demand (BOD), chemical oxygen demand (COD), total dissolved solids (TDS), total suspended solids (TSS), and conductivity were assessed according to the standard recommended methods [43].

The major common tanning heavy metal represented by Cr was measured before and after MgO-NPs treatment using atomic adsorption spectroscopy (A PerkinElmer Analyst 800 atomic spectrometer). The Cr heavy metal was detected by atomic absorption spectroscopy according to the absorption of light by free metallic ions.

### 2.9. Statistical Analysis

All results presented are the means of three independent replicates. Data were subjected to statistical analysis by a statistical package SPSS v.17. The mean difference comparison between the treatments was analyzed by *t*-test or the analysis of variance (ANOVA) and subsequently by Tukey HSD test at *p* < 0.05.

## 3. Results and Discussion

### 3.1. Isolation and Identification of Fungal Isolate

The fungal isolate E3 was isolated and underwent primary identification according to standard keys. Based on morphological and culturable characteristics, the fungal isolate E3 was belonging to *Rhizopus* sp., which was subjected to molecular identification based on amplification and sequencing of the internal transcribed spacer (ITS) gene. The sequence analysis revealed that the fungal strain E3 was highly related to *Rhizopus oryaze* (accession number: NR103596), with similarity percentages of 99%. Therefore, the fungal strain obtained in this study was specifically identified as *Rhizopus oryaze* strain E3 (Figure 1). The obtained ITS sequence was deposited in the gene bank under accession number MW774584.

*Rhizopus oryaze* are characterized by their platform’s highly secondary metabolites such as chemicals (fumaric acid, lactic acid, and ethanol), enzymes, fermentative compounds, and a wide range of by-products [44,45]. This wide range of metabolites increases the possibilities of *R. oryaze* to incorporate into various biomedical and biotechnological applications. To date, this is the first report that utilized *R. oryaze* as a biocatalyst for the green synthesis of MgO-NPs.

### 3.2. Myco-Synthesis of MgO-NPs

The myco-synthesis of metal and metal oxide NPs has recently drawn more attention due to high scalability, easy handling, high metabolite secretions. Moreover, fungi are characterized by high accumulation and high tolerance to metals, as well as high biomass production that is used to produce high active metabolites [46]. In this study, biomass filtrate of *R. oryaze* strain E3 was utilized as a biocatalyst for the green synthesis of MgO-NPs. The first monitor for successful fabrication was the color change of fungal biomass filtrate from colorless to turbid white after mixed with precursor (Mg (NO_3_)_2_.6H_2_O). Changing of the color indicates the activity of metabolites involved in biomass filtrate to reduce nitrate (NO_3_) to nitrite (NO_2_) and the liberated electron used to reduce Mg^2+^ to form Mg(OH)_2_, which was calcinated at 400 °C to form MgO-NPs (Equations (1) and (2)) [47].

To confirm MgO-NPs formation, the color change was monitored by UV-Vis spectroscopy to detect the maximum surface plasmon resonance (SPR). The size, shape, and well distribution of green synthesized NPs were usually influenced by SPR, as reported previously by Fedlheim and Foss [48]. In this regard, Jeevanandam et al. [49] reported that the size of biosynthesized MgO-NPs tends to be small when the SPR is less than 300 nm, whereas it became more anisotropic at SPR greater than 300 nm. In the present study, myco-synthesized MgO-NPs showed SPR at a wavelength of 282 nm (Figure 2), which indicates the presence of particles at the nanoscale, as reported previously [50]. The presented data are compatible with those reported by Abdallah et al. [51] and Nguyen et al. [50]; these studies showed that the maximum peaks of MgO-NPs synthesized by *Rosmarinus officinalis* L. and *Tecoma stans* (L.) were at 250 nm and 281 nm, respectively. Moreover, the chemically synthesized MgO-NPs showed a broad absorption peak at 290 nm [52]. Therefore, we can assume the efficacy of metabolites involved in biomass filtrate to reduce, cap, and stabilize MgO-NPs.

### 3.3. Optimizing Myco-Synthesis of MgO-NPs

Several environmental factors affect the biosynthesis processes, the stability of NPs, and their applications. These factors can affect the reducing agents such as enzymes, proteins, carbohydrates, and others present in biomass filtrate, which directly influence the biosynthesis processes [53]. The optimization of these factors has positive impacts on reducing and capping agents, decreasing the biosynthesis time, increasing NP stability, and decrease aggregation [32]. Among these factors, contact time, pH, temperature, and concentrations of precursor are investigated in the current study.

The incubation period of fungal biomass in distilled water to secrete bioactive compounds that acts as reducing and capping agents is considered a critical factor. In the present study, the optimum contacting time between fungal biomass and distilled water was 72 h. At this time, the color intensity drastically increased, referring to the efficacy of active compounds involved in biomass filtrate to reduce the highest amount of Mg(NO_3_)_2_·6H_2_O (Figure 3A). The concentration of Mg ions is considered an important factor that influences the myco-synthesis of MgO-NPs; therefore, different concentrations of precursor were used. In this study, the absorbance at λ_282_ was increased gradually by increasing Mg(NO_3_)_2_·6H_2_O concentration until 4 mM, after which the absorbance was decreased (Figure 3B). This indicates the efficacy of active metabolites including enzymes and protein to reduce Mg(NO_3_)_2_·6H_2_O, whereas the MgO-NPs tend to aggregate or agglomerate by increasing precursor concentrations and hence decreasing the absorbance [54]. Compatible with our study, Ibrahim [55] reported that the color intensity of Ag-NPs synthesized by banana peel extract was changed from yellowish-brown to dark reddish-brown by increasing the precursor (AgNO_3_) concentrations.

Another important factor affecting the myco-synthesis of MgO-NPs is the contact time or the reaction time between biomass filtrate and optimum precursor concentration (4 mM). The color changes to white precipitate offer evidence for the successful fabrication of MgO-NPs, which is monitored by detecting the absorbance at λ_282_. Our data showed that the highest absorbance was achieved at 36 h of reaction time between fungal biomass filtrate and Mg(NO_3_)_2_·6H_2_O by constant the previous optimum conditions (Figure 3C). At the early stages of reaction time, the SPR peak was broadened due to the slow reduction of metal ions to NPs. At the optimum reaction time, the large number of metal ions were reduced to NPs and identical SPR bands were formed. By increasing the reaction time, the absorbance gradually decreased due to the aggregation of some particles, consequently reducing the color intensity and particle size [55].

In addition, pH is considered an important factor that influences reducing agents, which affects the green synthesis. In the current study, different pH values were adjusted ranging from 6 to 11, and their impact on the reduction process was investigated through measuring the absorbance at λ_282_ after the constant of the other parameters. Data analysis showed that the alkaline solution was preferred for the myco-synthesis of MgO-NPs by *R. oryaze* strain E3. The highest absorbance at λ_282_ was attained at pH 8, which was evidence for maximum MgO-NPs production (Figure 3D). The lowest absorbance was attained at an acidic solution (pH = 6) with an absorbance value of 0.845 ± 0.01, which indicates that the activity of reducing groups present in biomass filtrate was reduced at acidic medium. Our recently published study showed that the activities of metabolites secreted by *Aspergillus carbonarious* strain D-1 were highly active at alkaline medium for reducing, cap, and stabilizing of α-Fe_2_O_3_-NPs and MgO-NPs [56]. The charges of biomolecules present in biomass filtrate can be altered due to differences in pH values and thus, the reducing capacity is affected [57].

Finally, the temperature is considered the main factor effect on the enzymes, proteins, carbohydrates, and other reducing agents present in biomass filtrate and consequently influence the NP synthesis process. In this study, different incubation temperatures (25–40 °C) were investigated to detect the preferable incubation temperature by measuring the absorbance intensity at λ_282_. Data analysis showed that the capacity of metabolites as reducing, capping, and stabilizing agents were accomplished at 35 °C (Figure 3E). The intensity of color as indicated by absorbance was decreased at a temperature greater or less than 35 °C. This phenomenon could be attributed to the inactivation of reducing agents at low and high incubation temperatures. Rai et al. [58] reported that the incubation temperature of the NP synthesis process can influence the nature, size, and shape of NPs formed. Interestingly, Patra and Baek [53] reported that the synthesis of NPs using physical methods needs an incubation temperature ˃ 350 °C, whereas chemical approaches require an incubation temperature ˂ 350 °C; in contrast, green approaches required ambient temperature.

### 3.4. Myco-Synthesized MgO-NPs Characterization

#### 3.4.1. Transmission Electron Microscopy

The morphological characteristics of myco-synthesized MgO-NPs including size and shape were investigated using TEM analysis. As shown in Figure 4A, the capability of metabolites secreted by *R. oryaze* strain E3 for reducing, capping, and stabilizing of well-dispersed spherical shape with sizes ranging from 8.0 to 47.5 nm with an average diameter of 20.38 ± 9.9 nm (Figure 4B). Recently, spherical MgO-NPs were successfully fabricated through harnessing metabolites of *Aspergillus carbonarious* D-1 with a size range of 20–80 nm [56]. Moreover, leaf aqueous extract of *Carica papaya* L. was used to synthesis of spherical MgO-NPs with an average size of 100 nm [59]. In this study, the obtained data evidence the potential for green synthesis of MgO-NPs with small size by harnessing metabolites of identified fungal strain. The activities of NPs are correlated to their size and shape, where the small sizes have more activities [36,60]. For instance, synthesized MgO-NPs with a varied size of 35.9, 47.3, and micron-size of 2145.9 nm exhibited bactericidal efficacy against *Bacillus subtilis* with percentages of 96.12%, 94.46%, and 75.71%, respectively [61]. The difference in the sizes and shapes of NPs synthesized by green approaches could be attributed to metabolites between different biological entities (bacteria, fungi, actinomycetes, algae, and plants), and even between the different species in the same genus [32].

#### 3.4.2. Scanning Electron Microscopy—Energy Dispersive X-ray (SEM-EDX)

The surface morphology of myco-synthesized MgO-NPs, agglomeration, and qualitative and quantitative chemical compositions were investigated using SEM-EDX analysis. The SEM image showed well-dispersed spherical MgO-NPs without any aggregation (Figure 4C). Furthermore, the EDX profile contains Mg and O with weight percentages of 19.3% and 7.9%, respectively, and with atomic percentages of 16.1% and 4.3%, respectively (Figure 4D). Moreover, the EDX profiles showed that the weight percentages of other elements present in the sample were 71.07%, 1.2%, and 0.6% for C, Cl, and Ca, respectively. Dobrucka [62] reported that the presence of peaks at an energy between 0.5 and 1.5 KeV indicates the successful synthesis of MgO-NPs, which is confirmed by our study. Consistent with the obtained data, the EDX profile of MgO-NPs synthesized by water extract of *Artemisia abrotanum* contains Mg, O, Al, Si, K, and Ca with weight percentages of 13.9%, 39.4%, 1.4%, 0.3%, 0.8%, and 0.5%, respectively [62]. The presence of peaks other than Mg and O could be attributed to the hydrolysis of capping and stabilizing agents such as proteins, enzymes, polysaccharides, and amino acids by X-ray [63].

#### 3.4.3. X-ray Diffraction (XRD) Analysis

The crystalline nature of myco-synthesized MgO-NPs was investigated using XRD analysis. Data showed five intense peaks at 2θ° of 36.9°, 42.6°, 62.2°, 75.4°, and 78.6° that correspond to planes of (111), (200), (220), (311), and (222), respectively (Figure 5A). These data confirmed that the as-formed MgO-NPs by *R. oryaze* strain E3 were crystallographic structures according to the JCPDS standard (JCPDS file No. 89-7746) [64]. The observed XRD peaks indicate the presence of oxide in the sample as Mg(OH)_2_ and MgO, confirmed by XPS analysis. Lekota et al. [65] reported that the peaks observed at 2θ° of 36.9° (111) and 75.4 (311) correspond to Mg(OH)_2_, whereas the diffraction peaks at 2θ° of 42.6° (200), 62.2° (220), and 78.6° (222) signified cubic MgO-NPs. The crystal size of myco-synthesized MgO-NPs was measured as 50 nm from the XRD pattern by the Debye–Scherrer equation.

#### 3.4.4. Dynamic Light Scattering (DLS)

The size distribution of MgO-NPs in colloidal solution was assessed using the DLS technique by reacting light beams with myco-synthesized MgO-NPs [66]. In this study, the average diameter size of MgO-NPs calculated using DLS was 56.1 nm, 60.5 nm, and 54.9 nm for volume intensities of 39.6%, 55.5%, and 4.9%, respectively, of colloidal solution (Figure 5B). In most cases, the size obtained from DLS is bigger than those obtained from TEM and XRD. This phenomenon can be attributed to the metabolites coating the NP surface which act as capping and stabilizing agents [63,66]. Moreover, the non-homogenous distribution of NPs in colloidal solution and hydrodynamic residue measured by DLS can give a bigger size [67,68].

Furthermore, the polydispersity index (PDI) that indicates the homogeneity of NPs in the colloidal solution can be measured based on DLS analysis. The homogeneity increases or decreases if the PDI values are below or higher than 0.4, while the colloidal sample is considered completely heterogenous if PDI values are higher than 1. The obtained data demonstrate that the PDI value of MgO-NPs synthesized by *R. oryaze* strain E3 was 0.2, which indicates the high homogenous colloidal solution.

#### 3.4.5. Fourier Transform Infrared (FT-IR) Spectroscopy

The bioactive compounds present in biomass filtrate of *R. oryaze* strain E3, which is responsible for the reduction of metal precursors to form MgO-NPs, are identified by FT-IR analysis which recorded a wavenumber between 400 to 4000 cm^–1^, as shown in Figure 6. The peak observed at 3700 cm^–1^ signifies to –OH stretching band [69]. The broadness peak at 3431 cm^–1^ is corresponding to the O–H stretching vibration mode of the hydroxyl groups overlapped with the NH stretching mode of amines [62]. The medium observed peaks at 1650 cm^–1^ are signified to the bending mode of primary amine (N–H) overlapped with either amide and carboxylate salt (see XPS analysis). The medium peak at 1435 cm^–1^ corresponds to the C=O stretching of carboxylate salt as well as the adsorption of CO_3_^2–^ and CO_2_ at the surface of MgO-NPs [70,71]. The adsorption of such functional groups on the surface of MgO-NPs has a critical role in catalytic reactions [72]. whereas the peak at 1030 cm^–1^ matched the Mg–OH stretching [73] with C-H out-of-plane bend. The peak located at 930 cm^–1^ corresponds to C-O stretching, *trans*-C-H out-of-plane bend, and P–O which refers to phosphate-containing molecules [74]. The successful fabrication of Mg-O was confirmed by peaks observed at a wavenumber between 400 to 700 cm^–1^, as reported in various published studies [69,75,76]. The peaks observed in FT-IR spectra reflect the role of metabolites involved in biomass filtration of *R. oryaze* strain E3 for reducing and stabilizing of MgO-NPs.

#### 3.4.6. X-ray Photoelectron Spectroscopy (XPS) Analysis

The XPS survey spectra of MgO-NPs synthesized by *R. oryaze* strain E3 is shown in Figure 7A. The magnesium was mainly characterized at different bending energies with different species Mg (1s, 2s, 2p, KL1, KL2, KL3 KL4, and KL5); other associated ions were detected as O (1s, 2s, KL1, and KL2), Cl (2p and 1s), and N 1s. The high-resolution spectra were performed for the most familiar peak, as shown below.

The nano-MgO included carbon in the main polysaccharide ring that was characterized by the presence of different peaks for O-C-O at 290.68 eV [77], O-C=O at 289.64 eV [78], N-C=O at 288.19 eV [79], while the other two peaks at 284.2 eV and 285.65 eV for C(C, N, H) and C(=H, O), C-O-C, respectively [79,80] (Figure 7B). This indicated the presence of amide and carbonyl groups in the polysaccharide’s skeletons (these results are parallel to the FT-IR analysis). Khan et al. reported that the presence of carbon labelled as C 1s in MgO-NPs could be attributed to the exposure of the sample to air before characterization [18]. The carboxyl group appeared in the O 1s (which split into two different peaks) at 532.12 eV, which is related to O-C=O, while the other peak is at 530.77 eV for O (N, C, H) [81,82] (Figure 7C). The N 1s deconvoluted into two peaks at 398.94 eV for N (C, H) [83] of the saccharide and 405.72 eV for NO_3_ originated from the precursor [84] (Figure 7D).

The HRES XPS spectra for Mg show a different types of species; the Mg 1s splitting into two peaks at 1304.24 eV and 1305.8 eV with At% 95.35% and 4.65% for MgO and Mg(OH)_2_, respectively [85], the Mg 2s has two different peaks at 87.97 eV and 89.21 eV with At% equivalent to 83.53% and 16.47% for both MgO and Mg(OH)_2_, respectively [86,87], while the Mg 2p shows 4 peaks at 49.22 eV and 50.08 eV with At% (i.e., 87.47% and 4.42% (major), respectively) for MgO, the other two peaks at 50.66 eV and 51.23 eV with At% ranged 6.18% and 1.92% (minor), respectively, for Mg-OH [87] (Figure 7E–G). This indicates that the majority of appearances of Mg is in MgO form with a trace amount of Mg(OH)_2_

### 3.5. Antimicrobial Activity

The antimicrobial activity of biosynthesized MgO-NPs was evaluated against different pathogenic Gram-positive and Gram-negative bacteria represented by *Staphylococcus aureus*, *Bacillus subtilis, Pseudomonas aeruginosa,* and *Escherichia coli* as well as unicellular fungi represented by *candida albicans*. The presented data showed that the activities of MgO-NPs against different pathogenic microbes were dose-dependent. Compatible with recently published studies, the activities of nanomaterials such as Ag, Fe_2_O_3_, Se, CuO, and ZnO were dose-dependent [88,89,90,91]. Analysis of variance showed that the zones of inhibitions (ZOIs) caused by 200 μg mL^–1^ were 14.7 ± 0.6, 14.3 ± 0.7, 13.7 ± 0.5, 10.6 ± 0.4, and 11.5 ± 0.5 mm for *C. albicans, E. coli, P. aeruginosa, S. aureus,* and *B. subtilis,* respectively (Figure 8). Similarly, the MgO-NPs fabricated by *Rhizophora lamarckii’s* extract have antibacterial activity against *Staphylococcus aureus, E. coli,* and *Streptococcus pneumoniae* and with a zone of inhibitions of 26.5, 26.1, and 26.3 mm, respectively [5].

The lowest concentration of MgO-NPs that inhibits microbial growth is defined as the minimum inhibitory concentration (MIC), which differs based on the microbial used. To detect the MIC value for each tested organism, different concentrations of MgO-NPs were investigated. Data analysis showed that the MIC value for *S. aureus* was 150 μg mL^–1^ with ZOI of 8.3 ± 0.6 mm, whereas *C. albicans, E. coli, P. aeruginosa,* and *B. subtilis* have MIC values of 100 μg mL^–1^ with ZOI of 9.7 ± 0.7, 9.0 ± 0.0, 9.8 ± 0.3, and 8.3 ± 0.6 mm, respectively (Figure 8).

The antimicrobial activity of biosynthesized MgO-NPs has been reported by several researchers [5,18,69]. These activities could be attributed to different mechanisms such as the enhancement of the production of reactive oxygen species (ROS), interactions between MgO-NPs and microbial cell walls, discharge of Mg^2+^ upon the entrance of microbial cells, and the alkaline effects of MgO-NPs on the microbial cells. Rai et al. [21] reported that the toxicity of nanoparticles synthesized using fungal species are dependent on size, shape, concentration, and surface charge of nanoparticles used.

The production of ROS, primarily superoxide radicals (^–^O_2_), hydrogen peroxide (H_2_O_2_), and reactive hydroxyl radicals (^•^OH), interferes with nucleic acids and proteins, ultimately leading to cell death [73]. In the current study, Gram-negative bacteria are more sensitive to biosynthesized MgO-NPs than Gram-positive bacteria and this phenomenon can be related to variations between two bacterial kinds in cell wall structures. The Gram-positive bacterial cell wall contains a thick layer of peptidoglycans; on the contrary, Gram-negative bacteria contain a thin layer of peptidoglycans and possess rich lipopolysaccharides (LPS). The interaction between NPs and bacterial cells is due to the negative charge of LPS and the positive charge of NPs [92,93]. Moreover, due to the thin peptidoglycan layer in Gram-negative bacteria, the MgO-NPs can penetrate the cell wall and deposit on the cell membrane, which changes the selective permeability function and is followed by cell death [94]. Furthermore, the MgO-NPs can disrupt the quorum sensing, which is responsible for communications between microbial strains, leading to the inhibition of microbial activities and functions [94,95].

The Mg^2+^ formed due to the penetration of MgO-NPs into the microbial cells can interact with thiol groups of amino acids, leading to the disruption of protein structure and ultimately cell death [75]. Sawai et al. [96] reported that a thin layer of water was adsorbed on the surface of MgO-NPs, resulting in a higher pH value in an aqueous solution than the equilibrium value, leading to microbial cell membrane damage upon contact.

### 3.6. Larvicidal Activity

The diseases caused by mosquitos are considered key challenges to public health worldwide and many efforts have been established to control these diseases. The conventional methods (either biological or chemical methods) have been utilized to control the spread of mosquitos, but these methods have drawbacks with effects such as toxicity to consumers and an increase in mosquito resistance to these compounds [97]. Recently, due to the unique characteristics of NPs, they have been used as an alternative source to control mosquito-borne diseases instead of conventional methods. To date, this is the first report to investigate the efficacy of myco-synthesized MgO-NPs against *C. Pipiens.*

The mosquitocidal mechanism of MgO-NPs may be related to two mechanisms: the production of ROS and lipid peroxidation and the leakage of internal cellular contents due to cell membrane damage [73,98]. Once MgO-NPs is sprayed over one stage of mosquito life cycles such as egg, larvae, pupa, or adult, it is dissociated into Mg^2+^ and O^2–^ ions in the surrounding environment. The high concentration of O^2–^ ions forms ROS, which ultimately leads to oxidative stress and lipid peroxidation. Moreover, the small size of MgO-NPs can react with nucleic acid and deform it, then inhibit the growth of mosquitos [99]. Moreover, the high concentration of Mg^2+^ leads to the destabilization or damage of cellular equilibrium, causing more stress, leakage of cellular components, and finally mosquito cell death [75] (Figure 9A).

In the current study, data analysis showed that the mortality percentages differed based on MgO-NPs concentrations. More than 50% mortality was observed in all concentrations after 72 h. The lowest mortality percentages (52%) were recorded after 3 days of treatment with 2 µg mL^–1^ of MgO-NPs, whereas the maximum mortality percentages (96%) were obtained at 10 µg mL^–1^. At MgO-NPs concentrations of 4, 6, and 8 µg mL^–1^, the mortality percentages reached 64%, 72%, and 84%, respectively, after 72 h (Figure 9B). The LC_50_ (concentration of MgO-NPs that inhibit 50% of the population) and LC_90_ (concentration of MgO-NPs that inhibit 90% of the population) were 2.21 µg mL^–1^ and 10.71 µg mL^–1^, respectively. Recently, several studies have reported the efficacy of different metal and metal oxides NPs to control or inhibit the mosquito populations [15,37,39]. The rod-shaped MgO/ hydroxyapatite nanoparticles showed potentiality to inhibit *Aedes aegypti, Anopheles stephensi,* and *Culex quinquefasciatus* through the accumulation of Mg^2+^, which destroys the mosquito cells [100].

### 3.7. Mosquito Repellent Activity

Data analysis showed that all tested concentrations exhibit more than 60% repellency against female *C. Pipiens* mosquitoes. However, the repellent activity of MgO-NPs was decreased with time increase (Figure 9C). The MgO-NPs concentrations of 10 mg/cm^2^, 5 mg/cm^2^, 2.5 mg/cm^2^, and DEET (as a positive control) showed repellency percentages of 95.16%, 90.34%, 87.45%, and 100%, respectively, for more than 60 min (Figure 9C). Interestingly, the repellency percentages between 10 mg/cm^2^ and the positive control do not show any difference after 240 min (94.1% and 94.2%) and 480 min (87.3% and 87.5%). After maximum time (8 h), the repellence percentages were 78.6%, 67.1%, and 46. 78% for MgO-NPs concentration of 5 mg/cm^2^, 2.5 mg/cm^2^, and 1.0 mg/cm^2^, respectively, as compared with DEET (87.6%). Based on the obtained data, the MgO-NPs have repellence activity at high concentrations as compared to commercial substances (EDDT). The TiO_2_-NPs exhibit repellence activity against *Aedes aegypti* with a value of 80.43% at 100 ppm [101]. The present study provides an eco-friendly, cost-effective, and simple approach to control the spread of *C. Pipiens* populations, as well as new repellent agents that can be used instead of commercial compounds.

### 3.8. Decolorization and Degradation of Tanning Effluents

The materials at the nanoscale are characterized by their eco-friendly and large surface area, and these features enable them to absorb many contaminants. The tanning wastewater is characterized by its greenish-blue color due to the extensive use of chrome ions and dyes during various processing steps [102]. Disposing of this effluent without treatment prevents the penetration of sunlight and hence decreases or hinders the oxidation of the pollutants [103]. Recently, global trends have been directed to reduce the chemical methods used for remediation and degradation of environmentally hazardous materials by newly advanced, eco-friendly, and safe methods. Hence, the efficacy of biosynthesized MgO-NPs as adsorbents to treat the tannery effluent was investigated at different concentrations (50, 75, and 100 µg/100 mL) and different contact times (60, 120, 180, and 240 min). Data analysis showed that the potentiality of MgO-NPs was dose- and time-dependent. The decolorization percentages were increased by increasing NP concentrations and contact time. This phenomenon could be attributed to the adsorption sites being increased by increasing the adsorbent concentrations [104]. The highest decolorizations were accomplished at 100 µg MgO-NPs after 180 min, a record of 95.6 ± 1.6% (Table 1). Analysis of variance showed that the difference between decolorization after 180 and 240 min is not significant and the time is considered a critical factor at a large scale. Therefore, the treatment of 100 mL tanning effluent with 100 µg of MgO-NPs for 180 min was selected as the optimum condition to study the physicochemical parameters of tanning effluents.

Tanning effluents are characterized by high contents of hazardous chemicals, chlorides, calcium phosphates, bicarbonates, sulfates, sodium, nitrates, potassium, and varied dissolved salts. Therefore, the physicochemical parameters including pH, TDS, TSS, BOD, COD, and conductivity are usually high [105]. Moreover, the physicochemical parameters of tanning effluents are varied based on tannery size, chemicals used according to the type of products, and the amount of water used [106]. Data presented in Table 2 show a high level of physicochemical parameters of crude tanning effluents before MgO-NPs treatment. The pH of crude tannery effluent is usually alkaline because of the usage of bicarbonates and carbonates [107]. Furthermore, the values of TSS, TDS, BOD, COD, and conductivity of crude tanning effluents are 8745.3 ± 5.5 mg L^–1^, 15,704 ± 4.1 mg L^–1^, 2355.7 ± 7.0 mg L^–1^, 651.7 ± 4.7 mg L^–1^, and 26,738.7 ± 6.0 S m^–1^, respectively (Table 2). The high values of TSS and TDS may be attributed to the high concentrations of salts and inorganic or organic insoluble substances, which make tanning effluents unsuitable for plant irrigations [108]. The high level of electrical conductivity can be attributed to the high usage of acid and salts such as chromium salts and sodium during various tanning processes [109]. The high level of BOD and COD of untreated effluents have negative impacts on aquatic and environmental ecosystems [110]. Data analysis showed that the MgO-NPs have high efficiency to highly reduce the physicochemical parameters of tanning effluents. As shown in Table 2, the removal percentages of TSS, TDS, BOD, COD, and conductivity due to MgO-NPs treatment were 97.9%, 98.2%, 87.8%, 95.9%, and 97.3%, respectively. This activity can be due to high MgO-NPs concentration, which provides the large adsorption sites and large surface area that facilitate the adsorption of most organic ions and other pollutants, especially at high contact times [111].

Heavy metals are considered one of the most damaging environmental pollutants and can cause severe human and animal problems due to their accumulation in the food chains [112]. The tanning industry is one of the main sources of the discharge of heavy metals into the environment. Among these heavy metals is chromium ions, which, despite their presence in some daily diets, can cause skin allergies and lung cancer [113]. MgO-NPs have various advantages to be used as a bio-adsorbent for different heavy metals, such as cost-effectiveness, high adsorption capacity, nontoxicity, eco-friendly, abundance, and biocompatibility [17]. In the current study, data analysis showed the efficacy of biosynthesized MgO-NPs to reduce chromium ion in tanning effluents from 822.3 ± 2.5 mg L^–1^ to 14.5 ± 0.9 mg L^–1^ with removal percentages of 98.2%. This high removal efficacy can be attributed to the precipitation and adsorption of metals on the MgO; on the contrary, for other nanomaterials such as Al_2_O_3_ and TiO_2_, the removal mechanism was adsorption only as mentioned previously [114]. Yang et al. [115] reported that the high adsorption efficacy of MgO-NPs is due to the dissociation of OH^–^ from Mg(OH)_2_, along with the synergistic effects of the adsorption and precipitation process. Interestingly, MgO-NPs showed the adsorption efficacy of Pb (II) and Cd (II) with values of 2614 and 2294 mg g^–1^ [116], respectively. Recently, the MgO-NPs synthesized by *Aspergillus niger* F1 showed the removal of Cr with percentages of 94.2 ± 1.2% [76]. Moreover, Seif et al. reported the high potentiality of MgO-NPs to adsorb Cr^3+^ with values of 1033.8 mg g^–1^ as compared to montmorillonite nanoparticles, for which the maximum adsorption was 3.6 mg g^–1^ [117].

## 4. Conclusions

In this study, synthesis of MgO-NPs was pursued by using the metabolites on the biomass filtrate of *Rhizopus oryaze* with Mg (NO_3_)_2_.6H_2_O as a precursor. The production technique was optimized by studying the precursor concentration, the contact time between the fungal biomass filtrate and precursor, incubation temperatures, and pH values. The color change to turbid white and the maximum SPR at 282 nm confirmed the successful synthesis of MgO-NPs, which is characterized by TEM, SEM-EDX, XRD, DLS, FT-IR spectroscopy, and XPS analyses. These analyses revealed that the biosynthesized MgO-NPs are crystalline, spherical, and well-dispersed, with sizes ranging between 8.0 to 47.5 nm. The fungal-induced MgO-NPs offer potential anti-microbial activity against *S. aureus*, *B. subtilis, P. aeruginosa, E. coli,* and *C. albicans,* with varied inhibition zones based on NP concentrations. Moreover, these NPs provide a modest, eco-friendly, and cost-effective way of controlling *C*. *Pipiens* populations with enhanced repellent activity as compared with commercial compounds. Additionally, they could be used efficiently to decolorize tanning effluents through the reduction of physicochemical parameters such as TSS, TDS, BOD, COD, and conductivity. Moreover, the biosynthesized MgO-NPs exhibit high efficacy to bio-adsorb chromium ions from tanning effluents. This study provides a simple, eco-friendly, cost-effective, and rapid approach to inhibit the growth of the microbial pathogen, prevent the spread of adverse insects, treat some of the worst environmental contaminants, and adsorb the most hazardous heavy metal.

## Figures and Tables

**Figure 1 jof-07-00372-f001:**
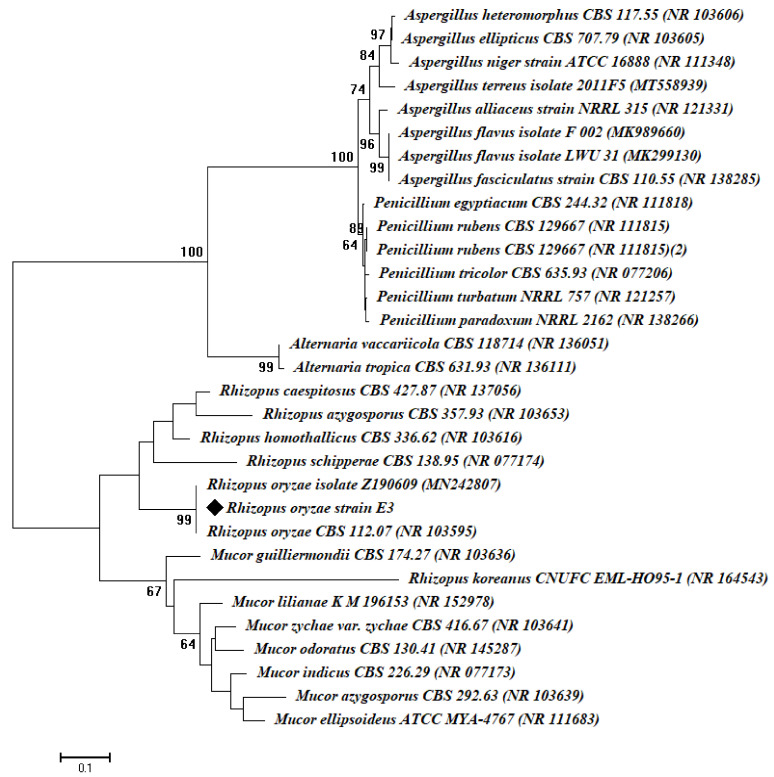
Phylogenetic tree of the fungal strain E3 with the sequences from NCBI. The symbol ♦ refers to ITS fragments retrieved from this study. The tree was conducted with MEGA 6.1 using the neighbor-joining method.

**Figure 2 jof-07-00372-f002:**
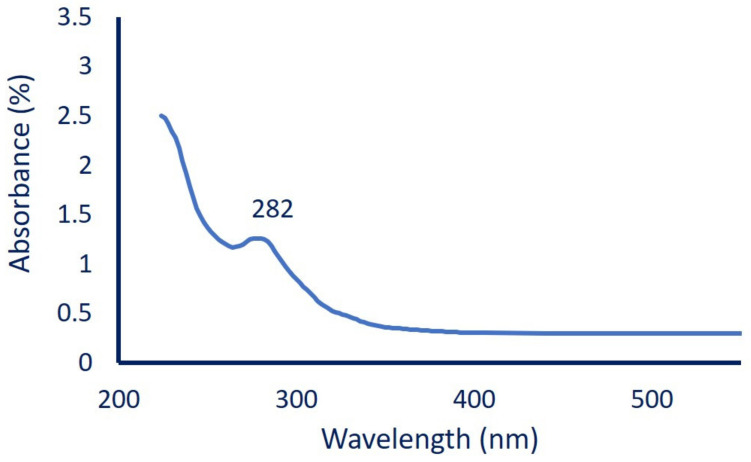
UV-Vis spectroscopy of myco-synthesized MgO-NPs showed maximum SPR at 282 nm.

**Figure 3 jof-07-00372-f003:**
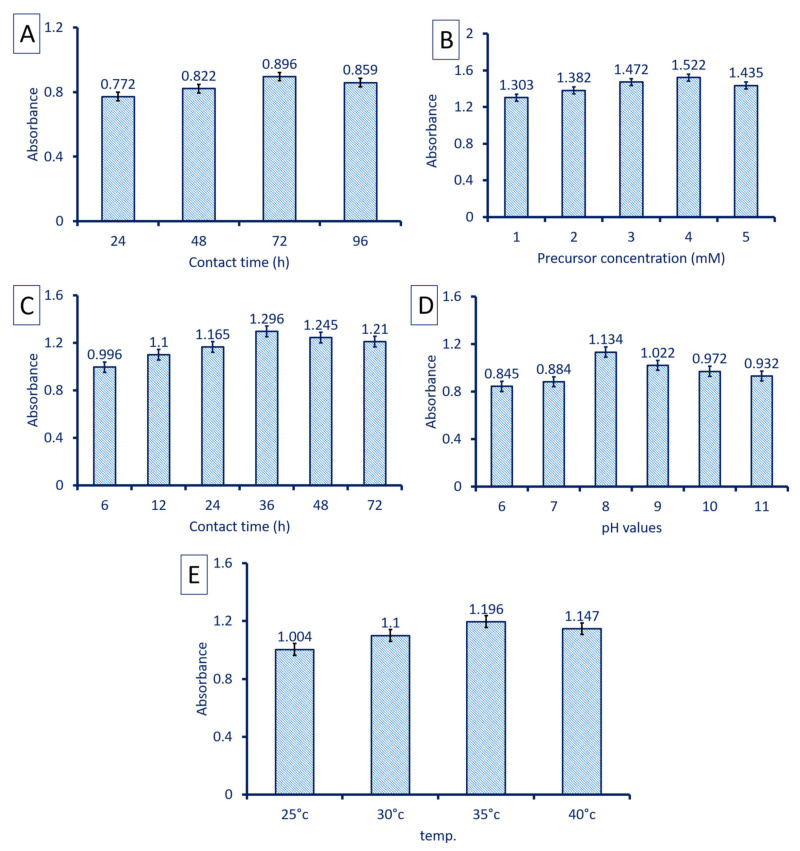
Optimizing factors for myco-synthesis MgO-NPs using *R. oryaze* strain E3. (**A**) is contact time between fungal biomass and distilled water, (**B**) is precursor concentrations, (**C**) is contact time between biomass filtrate and optimum precursor concentration, (**D**) is the effect of pH values, and (**E**) is the effect of incubation temperature on the biosynthesis process.

**Figure 4 jof-07-00372-f004:**
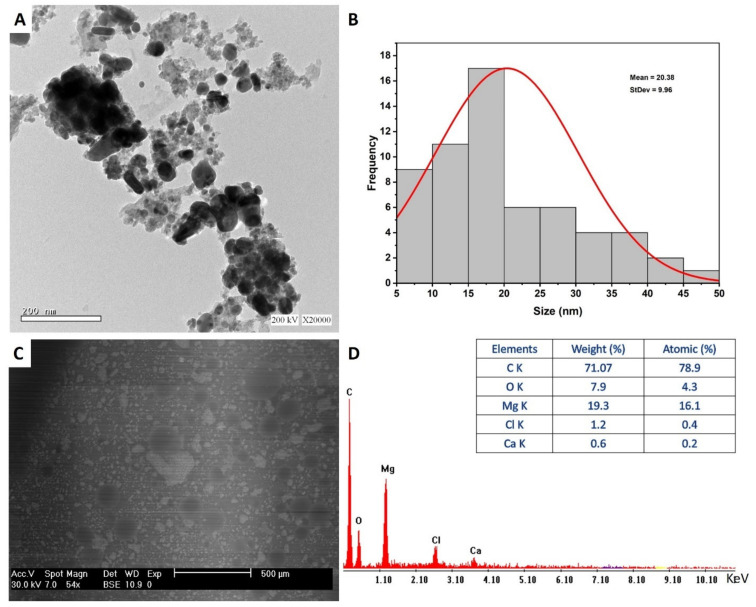
Characterization of myco-synthesized MgO-NPs. (**A**) TEM image, (**B**) size distribution, (**C**) SEM image, and (**D**) EDX spectrum.

**Figure 5 jof-07-00372-f005:**
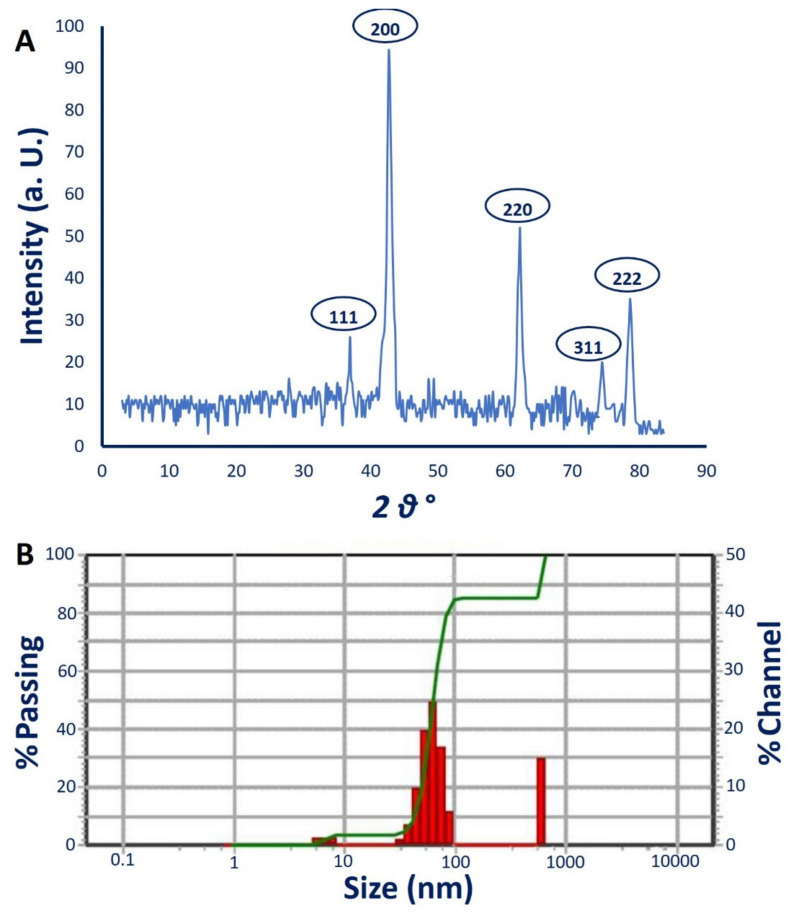
(**A**) XRD analysis showed the crystalline nature; (**B**) DLS analysis of myco-synthesized MgO-NPs.

**Figure 6 jof-07-00372-f006:**
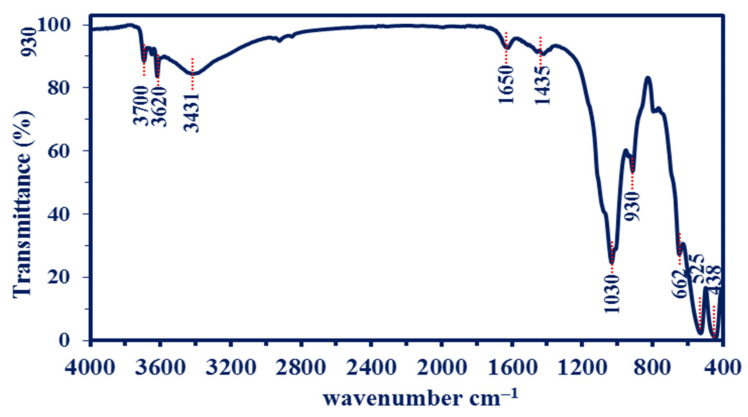
The FT-IR spectrum of myco-synthesized MgO-NPs fabricated by metabolites of *R. oryaze* strain E3.

**Figure 7 jof-07-00372-f007:**
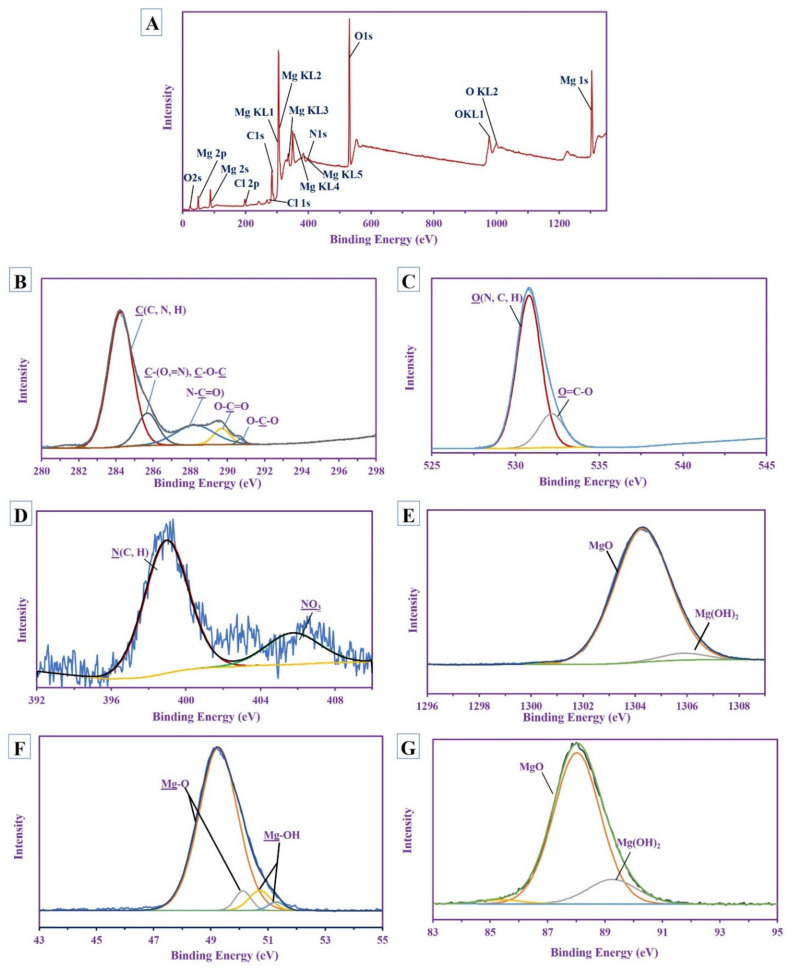
The X-ray photoelectron spectroscopy (XPS) analysis of biosynthesized MgO-NPs. (**A**) Overall view; (**B**) C 1s; (**C**) O 1s; (**D**) N 1s; (**E**) Mg 1s; (**F**) Mg 2p; (**G**) Mg 2s.

**Figure 8 jof-07-00372-f008:**
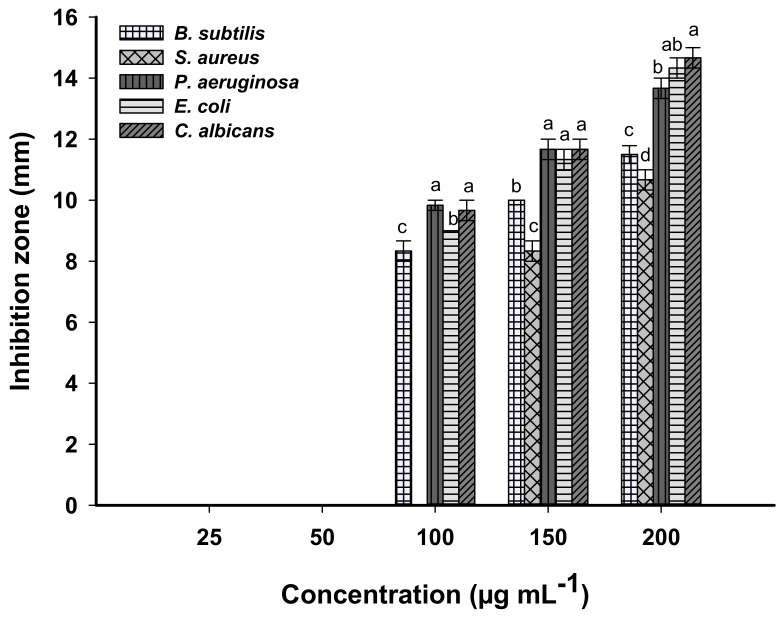
The antimicrobial activity of MgO-NPs at different concentrations against Gram-positive and Gram-negative bacteria, and unicellular fungi. Different letters (a, b, c, and d) on bars at the same concertation denote that mean values are significantly different (*p* ≤ 0.05) (*n* = 3).

**Figure 9 jof-07-00372-f009:**
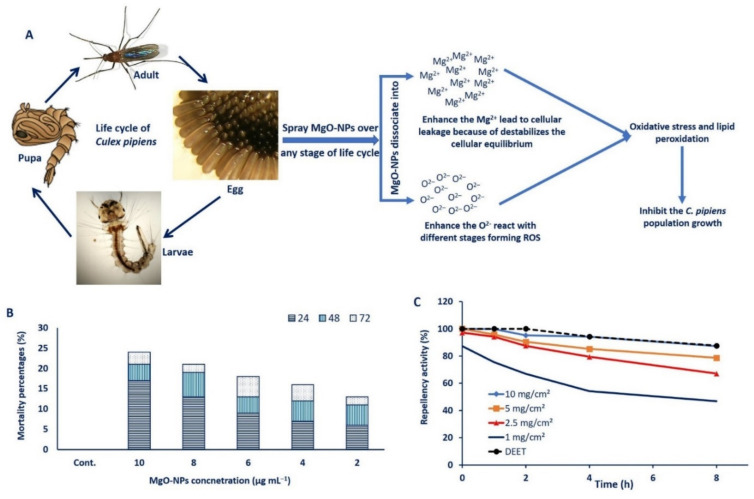
Mosquitocidal activity of MgO-NPs against *C. Pipiens.* (**A**) Proposal mechanism of MgO-NPs as mosquitocidal; (**B**) larvicidal activity of MgO-NPs against *C. Pipiens* at different concentrations (2, 4, 6, 8, and 10 µg mL^−1^) after different contact times (24, 48, and 72 h); (**C**) repellence activity of different concentrations of MgO-NPs as compared to a positive control (DEET).

**Table 1 jof-07-00372-t001:** Decolorization percentages (%) of tanning effluents using different concentrations (50, 75, and 100 µg/100 mL) of myco-synthesized MgO-NPs at different contact times (60, 120, 180, and 240 min).

MgO-NPs Concentration /100 mL Tanning Effluent	Decolorization Percentages (%) after Time (min)			
	60 min	120 min	180 min	240 min
Control	2.3 ± 0.2 ^a^	3.5 ± 0.4 ^b^	4.6 ± 0.3 ^c^	5.4 ± 0.4 ^c^
50 µg	34.3 ± 2.2 ^a^	48.7 ± 3.7 ^b^	54.2 ± 2.02 ^c^	59.6 ± 2.5 ^c^
75 µg	44.5 ± 3.3 ^a^	57.7 ± 2.4 ^b^	71.3 ± 2.3 ^c^	75.2 ± 1.7 ^c^
100 µg	65.4 ± 1.9 ^a^	81.1 ± 1.6 ^b^	95.6 ± 1.6 ^c^	96.7 ± 0.7 ^c^

Different letters in the same row are significantly different (*p* ≤ 0.05) by the Tukey LSD test. Data are represented by mean ± SD (*n* = 3).

**Table 2 jof-07-00372-t002:** Physicochemical characterizations and chromium ion adsorption from tanning effluents by MgO-NPs.

Physicochemical Parameters	Control	After MgO-NPs Treatment	Removal Percentages (%)
pH	10.5	8	-
TSS (mg L^–1^)	8745.3 ± 5.5 ^a^	177.7 ± 5.1 ^b^	97.9
TDS (mg L^–1^)	15,704 ± 4.1 ^a^	286.7 ± 4.2 ^b^	98.2
BOD (mg L^–1^)	2355.7 ± 7.0 ^a^	287.3 ± 4.9 ^b^	87.8
COD (mg L^–1^)	651.7 ± 4.7 ^a^	26.7 ± 2.1 ^b^	95.9
Conductivity (S m^–1^)	26,738.7 ± 6.0 ^a^	708.7 ± 4.0 ^b^	97.3
Cr (mg L^–1^)	822.3 ± 2.5 ^a^	14.5 ± 0.9 ^b^	98.2

Different letters in the same row are significantly different (*p* ≤ 0.05) by the Tukey LSD test. Data are represented by mean ± SD (*n* = 3).

## Data Availability

The data presented in this study are available on request from the corresponding author.

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
