# Peer review of "Rhizopus oryzae-Mediated Green Synthesis of Magnesium Oxide Nanoparticles (MgO-NPs): A Promising Tool for Antimicrobial, Mosquitocidal Action, and Tanning Effluent Treatment"

_jof, 2021, doi:10.3390/jof7050372_

Round 1
Reviewer 1 Report
Attached

Author Response
Thank you very much for reviewing the manuscript. we answer all reviewer comments point by point. Please see the attachment
Response to Reviewers Report-1
(jof-1203647)
To Reviewer #1
Thank you very much for reviewing our manuscript, your agreement, and valuable comments. We are also grateful for the favorable comments. The manuscript was revised by native speaker whose English is the native speaker. We made corrections and we hope they meet with your approval. The detailed explanation is given below.
Reviewer comment #: I have gone through the manuscript “Rhizopus oryzae-mediated green synthesis of magnesium oxide nanoparticles (MgO-NPs): a promising tool for used as antimicrobial, mosquitocidal agent, and tanning effluent treatment” thoroughly. The theme of presented MS is interesting and author used recent studies. Overall MS content is good and I would be happy to recommend this MS to be accepted for final publication with minor corrections as follows.
Author response #: Thank you very much for your agreement and your comment. we made the correction point by point as shown below.
Reviewer comment #: The title of the manuscript should be reframed like Rhizopus oryzae-mediated green synthesis of magnesium oxide nanoparticles (MgO-NPs): A promising tool for antimicrobial, mosquitocidal action and tanning effluent treatment or Title need to be modified for clear meaning.
Author response #: Thank you very much for your comment. the title was revised as reviewer recommend.
Reviewer comment #: In abstract: Line 28 Sentence should start with capital T in The.
Author response #: Thank you for your correction. Done
Reviewer comment #: In abstract: Line 30 C should be capital in candida albicans.
Author response #: Thank you for your comment. Done
Reviewer comment #:In abstract: Line 34 Tannery effluents should be there instead of tanning effluent. In keywords also, use tannery effluents instead of tanning effluent.
Author response #: Thank you for your correction. Done
Reviewer comment #: Line 45-47: Reframe it as it is incomplete. Among these new characters…….size to surface area are some of the essential features.
Author response #: Thank you for your comment. The sentence was revised as follows: “Nanoscience means the production of new materials at the nanoscale (1-100 nanometers). The prepared materials have unique properties that are not found in bulk materials [1]. Among these unique properties; shape, size, compatibility, surface charge, chemical stability, catalytic activity, and small size to a large surface area [2].”
Reviewer comment #: Line 52: Write took place instead of take placed.
Author response #: Thank you for your correction. Done
Reviewer comment #: Line 55: Correct the sentence to: The disadvantages associated with physical and chemical methods.
Author response #: Thank you for your correction. Done
Reviewer comment #: Line 57: Write Ecofriendly nature instead of ecofriendly. The entire sentence should be reframed.
Author response #: Thank you for your comment. the sentence was revised as follows: “The biological synthesis or green synthesis has been describing as cost-effectiveness, biocompatibility, eco-friendly nature, scalability, avoid harsh conditions, and unutilized of hazardous chemicals.”
Reviewer comment #: Line 62: Write Metal oxide nanoparticles instead of metal oxide nanoparticle.
Author response #: Thank you for your correction. Done
Reviewer comment #: Line 67: Write past two decades instead of two past decades.
Author response #: Thank you for your correction. Done
Reviewer comment #: Line 70: In Biological, write small b.
Author response #: Thank you for your correction. Done
Reviewer comment #: Line 81: Write which affects myco-synthesis of instead of which effects on myco-synthesis.
Author response #: Thank you for your correction. Done
Reviewer comment #: Line 102: The appeared colonies. Write capital T.
Author response #: Thank you for your correction. Done
Reviewer comment #: Line 139: Make it clearer ‘‘were running with the experiment’’.
Author response #: Thank you for your comment. the sentence was revised as follows: “The controls including fungal biomass filtrate and Mg(NO3)2·6H2O solution were running alongside the experiment under the same conditions.”
Reviewer comment #: Line 157: and underwent vaccum dessication instead of undergo.
Author response #: Thank you for your correction. Done
Reviewer comment #: Line 175: Write as ‘with KBR pellets, pressure was applied to form a disk and scanning was done in the range of 400-4000 cm-1.
Author response #: Thank you for your comment. Done
Reviewer comment #: Line 182: Write Passed energies instead of pass energy.
Author response #: Thank you for your correction. Done
Reviewer comment #: All the headings style should be uniform. Either put dot after all of them or not like2-4. Mosquitocidal assay. No dot is there.
Author response #: Thank you for your comment. All headings are uniformed as show in revised manuscript.
Reviewer comment #: Line no. 195 and 196: Use capital T in The.
Author response #: Thank you for your correction. Done
Reviewer comment #: Overall language is very complex. It needs to be simplified.
I would suggest improve the grammar portion too.
Author response #: Thank you for your comment. the manuscript was revised by native speaker whose English is the native language.
Reviewer comment #: Line no. 204 and 205: Write like this Fish was added to each tray for two weeks for optimum feeding of the larvae as follows:
Author response #: Thank you for your correction. Done
Reviewer comment #: Line no. 207: WRITE appeared instead of appear.
Author response #: Thank you for your correction. Done
Reviewer comment #: Line no. 245: Write Ta and Tb denotes instead of Ta and Tb are denoting.
Author response #: Thank you for your correction. Done
Reviewer comment #: Line no. 250: Write capital T in The.
Author response #: Thank you for your comment. Done
Reviewer comment #: Line no. 252: Write ‘to reach’ instead of ‘to reached’.
Author response #: Thank you for your correction. Done
Reviewer comment #: Line no. 254: Write ‘and its optical density was measured’.
Author response #: Thank you for your correction. Done
Reviewer comment #: Line no. 294: Write like this. Myco-synthesis of metal and metal oxide NPs has recently taken more attention due to.
Author response #: Thank you for your correction. Done
Reviewer comment #: Line no. 303: Write: which was calcinated instead of which calcined.
Author response #: Thank you for your correction. Done
Reviewer comment #: Line no. 304: Write : To confirm.
Author response #: Thank you for your correction. Done
Reviewer comment #: Be uniform in writing refence. Either put dot after et al. in all or not. (line no.307 and 312).
Author response #: Thank you for your correction. It is standardized as follows: “et al.”
Finally, we hope the response meets the reviewer approve.

Reviewer 2 Report
The antimicrobial activity of nanoparticles are made with agar diffusion test.
The authors report the MIC values using the agar diffusion test using different concentration of nanoparticle.
This assay is not determine the Minimum Inhibition concentration but the inhibition of the growth at that concentration.
to determine the MIC value must be used a different assay.
it is not correct to use the same medium to growth bacteria and fungi.
Mueller hilton is a medium to growth bacteria essentially .
the antimicrobial activity to determined the values MIC must be made with appropiate methods and medium.
Author Response
Thank you very much for reviewing the manuscript. we answer all reviewer comments point by point. Please see the attachment
Response to Reviewers Report-2
(jof-1203647)
Thank you very much for reviewing our manuscript. We are also grateful for your valuable comments. A detailed explanation is below.
Reviewer comment #: The antimicrobial activity of nanoparticles are made with agar diffusion test.
The authors report the MIC values using the agar diffusion test using different concentration of nanoparticle.
This assay is not determine the Minimum Inhibition concentration but the inhibition of the growth at that concentration.
to determine the MIC value must be used a different assay.
it is not correct to use the same medium to growth bacteria and fungi.
Mueller hilton is a medium to growth bacteria essentially.
the antimicrobial activity to determine the values MIC must be made with appropiate methods and medium.
Author response #: thank you for your comment. Really, there are other different methods used to investigate antimicrobial activity and detect the MIC, while the agar diffusion method is standard method and used to assessed antimicrobial activity of nanoparticles in various published study such as,
- Antibacterial Activity of TiO2- and ZnO-Decorated with Silver Nanoparticles. Compos. Sci.2019, 3(2), 61; https://doi.org/10.3390/jcs3020061.
- Characterization and the evaluation of antimicrobial activities of silvernanoparticles biosynthesized from Carya illinoinensis leaf extract. Heliyon 6 (2020) e036242. https://doi.org/10.1016/j.heliyon.2020.e03624.
Regarding to the growing media, please accept my apology for this mistake, I revised it after checking the raw data. we subculture the microbial strains on specific media at the first, after that we used Muller Hinton to assess the antimicrobial activity (for bacteria strains) and YEPD agar media (for Candida albicans). We clarify this meaning in revised manuscript as follows: “The bacterial strains were subcultured on nutrient agar media (containing g L–1: peptone, 5; Beef extract, 3; NaCl, 5; Agar, 20; distilled water, 1000 mL ), while C. albicans was subculture on yeast extract peptone dextrose (YEPD) agar media (containing g L–1: glucose, 20; peptone, 20; yeast extract, 10; Agar, 20; distilled water, 1000 mL) for 24 h. To check antimicrobial activity, each strain was homogenously streaked over Muller Hinton agar (for bacterial strains) and YEPD agar plates (for C. albicans) using a sterilized cotton swab.”.
The MIC values were detected according to the lowest concentration of MgO-NPs can inhibit the microbial growth (Based on zone of inhibition) and this method are published in various study. Actually, there are other methods used to detect MIC such as micro titer plate, in the current time we cannot repeat this experiment using other method due to Covid-19 pandemic spread and we will take in consideration in the future study.
Finally, we hope the response meets the reviewer approve.

Round 2
Reviewer 2 Report
The manuscript can be publish in this form.
Author Response
Thank you very much for your agreement.